# The Effect of *Fusarium verticillioides* Fumonisins on Fatty Acids, Sphingolipids, and Oxylipins in Maize Germlings

**DOI:** 10.3390/ijms22052435

**Published:** 2021-02-28

**Authors:** Marzia Beccaccioli, Manuel Salustri, Valeria Scala, Matteo Ludovici, Andrea Cacciotti, Simone D’Angeli, Daren W. Brown, Massimo Reverberi

**Affiliations:** 1Department of Environmental Biology, University of Rome “Sapienza”, 00185 Rome, Italy; marzia.beccaccioli@uniroma1.it (M.B.); manuel.salustri@uniroma1.it (M.S.); ludovicim@hotmail.it (M.L.); andrea.cacciotti@uniroma1.it (A.C.); simone.dangeli@uniroma1.it (S.D.); massimo.reverberi@uniroma1.it (M.R.); 2Centro di Ricerca Difesa e Certificazione, Consiglio per la Ricerca in Agricoltura e l’Analisi dell’Economia Agraria, 00156 Rome, Italy; 3United States Department of Agriculture, Agriculture Research Service, Peoria, IL 61604, USA; Daren.Brown@ars.usda.gov

**Keywords:** lipids, sphingolipids, mycotoxins, programmed cell death, necrotroph, salicylic acid

## Abstract

*Fusarium verticillioides* causes multiple diseases of *Zea mays* (maize) including ear and seedling rots, contaminates seeds and seed products worldwide with toxic chemicals called fumonisins. The role of fumonisins in disease is unclear because, although they are not required for ear rot, they are required for seedling diseases. Disease symptoms may be due to the ability of fumonisins to inhibit ceramide synthase activity, the expected cause of lipids (fatty acids, oxylipins, and sphingolipids) alteration in infected plants. In this study, we explored the impact of fumonisins on fatty acid, oxylipin, and sphingolipid levels *in planta* and how these changes affect *F. verticillioides* growth in maize. The identity and levels of principal fatty acids, oxylipins, and over 50 sphingolipids were evaluated by chromatography followed by mass spectrometry in maize infected with an *F. verticillioides* fumonisin-producing wild-type strain and a fumonisin-deficient mutant, after different periods of growth. Plant hormones associated with defense responses, i.e., salicylic and jasmonic acid, were also evaluated. We suggest that fumonisins produced by *F. verticillioides* alter maize lipid metabolism, which help switch fungal growth from a relatively harmless endophyte to a destructive necrotroph.

## 1. Introduction

Fungi derive nutrients from plants passively as biotrophs, aggressively as necrotrophs, or as hemibiotrophs in which the fungus first establishes itself in host tissue causing little damage followed by a switch to a destructive, necrotrophic lifestyle [1]. Although such classifications are useful, they are simplistic and somewhat arbitrary. Indeed, some fungi have been associated with all three growth strategies [2]. For example, *Fusarium verticillioides* is considered a hemibiotroph, co-exists with *Zea mays* (maize) as an endophyte while, on occasion, it is able to switch to a necrotroph and cause disease at any life stage of the plant [3,4].

What triggers this infrequent change in lifestyle is unknown, although it may be related to drought or heat stress or physical damage caused by insects or hail [5]. A frequent co-occurrence with pathogenic growth in infected maize is the production of fumonisins (FBs), a family of highly toxic secondary metabolites. Consumption of FB contaminated foods and feed is associated with a variety of animal diseases and is epidemiologically associated with esophageal cancer in humans [6,7,8,9].

The co-occurrence of disease and FBs suggest that they are associated. An early study found that a non-producing fumonisin strain of *F. verticillioides* caused ear infection and ear rot of maize, indicating that they are not related [10]; more recent studies support a role for FBs in pathogenesis. For example, in *F. verticillioides*, a mutation in FUM1, a polyketide synthase necessary for fumonisins production, generate a mutant less virulent to maize seedlings [11]; in *F. musae*, a strain enabled to synthesize FBs became pathogenic to seedlings [12,13]; and in *F. proliferatum*, fumonisin-deficient strains display significantly decreased pathogenicity to rice [14]. The impact of FBs on disease is likely due to their ability to inhibit ceramide synthase (CerS), a key enzyme in sphingolipid biosynthesis. Could the changes in plant metabolism of lipids caused by FBs trigger the fungus to transition from endophytism to necrotrophy? Sphingolipids are a diverse group of compounds that function as anchors for membrane proteins [15] and as secondary messengers for multiple cellular functions [16,17]. CerS forms an amide bond between the amine group on the sphingoid base, referred to as a long-chain base (LCB), and the carboxylic group of a fatty acyl-CoA [18]. Fatty acids with between 12–18 carbons (C12–C18) are called long-chain fatty acids (LCFAs) and chains between 20–36 carbon (C20–C36) are called very long-chain fatty acids (VLCFAs). Both the LCB and the fatty-acid chain can be further modified leading to even more sphingolipid structural diversity.

Perturbations in sphingolipid metabolism and programmed cell death (PCD) in plants are linked [19]. FBs, the structurally related fungal chemical AAL toxin, produced by *Alternaria alternata*, sphingolipid metabolism inhibitors induce the accumulation of LCBs, which in turn cause PCD [7,20]. PCD is dependent on the hormones ethylene, jasmonic acid (JA), and salicylic acid (SA) [21,22,23,24]. PCD may also be triggered by changes in ceramide levels. *A. thaliana* mutant *acd5* (encoding a ceramide kinase) accumulated ceramides prior to PCD, which was attenuated by the addition of phosphate ceramides [25]. The ceramide-triggered PCD may, in turn, be influenced by fatty acid chain length. Depletion of VLCFA ceramides can impair plasma membrane organization leading to reduced pathogenic responses and cell death [26,27]. 

The role of hormones (i.e., SA and JA) in helping fungi resist PCD caused by biotrophic and hemibiotrophic pathogens is crucial because they can induce production of defense proteins and other chemicals. For example, SA may induce production of the antifungal protein (PR1) as well as different enzymes involved in phytoalexin synthesis such as phenylalanine ammonia-lyase (PAL). Phenols, including the phytoalexins, are key molecules in the primary inducible response in plant defense. JA, which is produced in response to the necrotrophic pathogens, may induce the expression of the chitinase PR4 as well as several enzymes that promote lipid peroxidation including lipoxygenases (LOXs) and allene oxide synthases (AOSs). The cross-talk between SA and JA provide plants, through PCD, tools to restrict the growth of biotrophic and necrotrophic pathogens and limit their negative impact [28,29,30].

Fatty acids (FAs) may regulate the signal transduction as a consequence of environmental and developmental stimuli. FAs are play important roles in the membrane composition and fluidity as well as intracellular signaling which impact the plants immune responses [31]. Length, desaturation grade, and quantity are important parameters to determine the cellular response during plant–microbe interactions. For example, fatty acid synthesis can depend on the presence of palmitoleic acid (16:1) produced by the plant during the interaction of arbuscular–mycorrhizal fungi [32]. Furthermore, enhanced levels of palmitoleic acid (16:1) can confer resistance to pathogenic fungi [33]. Seed fatty acid composition is also suggested to be a component of pathogen susceptibility and seed colonization; in fact, reduced levels of oleic acid (18:1) results in the constitutive activation of defense responses [31,32].

Among the membrane lipids are the polyunsaturated FAs (PUFAs), which may be released in response to a pathogen attack. PUFAs may act as free FAs or as oxylipins or oxygenated FAs. During the infection process, initial plant defenses are activated by the detection of reactive oxygen species which promote lipid peroxidation and the formation of oxylipins. Among the oxylipins, JA elicits the plant defenses against biotrophic pathogens. Oxylipins are also produced by the fungi, influencing mycelial growth and spore germination. In *Aspergillus* spp., linoleic acid (18:2) influences development, seed colonization, and mycotoxin production [33,34].

In the present study, we evaluated the abundance of more than 50 sphingolipids including sphingoid bases, ceramides, and phytoceramides, in maize infected with an FB-producing *F. verticillioides* (wild-type) and an FB non-producing mutant (*fum1*∆), in which the *fum1* gene was deleted. We found that VLCFA containing sphingolipids were negatively impacted by FBs produced by the wild-type strain in infected maize while LCFA containing sphingolipids were positively impacted, compared to *fum1*∆ mutant infected maize. Lipid content can explain the pathogenic efficacy [17]; therefore, the fatty acids and oxylipins have been evaluated for the regulatory role during plant–pathogen interaction. We also evaluated the production of SA and JA in maize. We suggest that during the interaction between the host and *F. verticillioides*, the modification of sphingolipid and plant defense responses are associated with the fumonisin production and modulate the endophytic growth of *F. verticillioides* into a necrotrophic growth typical of rotting diseases. 

## 2. Results

### 2.1. Infection of Maize Kernel with F. verticillioides Wild-Type and Fumonisin Non-Producing Mutant fum1∆ 

Maize kernels infected with the *F. verticillioides* wild-type or mutant *fum1*∆ was maximal at 14 days after infection (dai) (Figure 1a–f). To understand the influence of fungal growth on seed germination, the length of roots and stems was evaluated. However, no significant difference was observed in root and stem length in 14-day seedlings infected (Figure 2a).

Consistent with previous work [35] showing that *F. verticillioides* may cause necrotic lesion [35], we observed more necrotic lesions in wild-type infected roots than in *fum1*∆ infected roots (Figure 1d and f and Figure 2b). The differences in disease symptoms that we saw did not appear to be related to the amount of pathogen presence, as determined by qPCR. The amount of genomic DNA detected was higher for *fum1*∆ than the wild-type infected maize (Figure 2c). 

### 2.2. Fumonisin Contamination and Lipid Compounds 

Lipid entities present during the maize kernel infected with the *F. verticillioides* wild-type or *fum1*∆ mutant were evaluated. Correlation between lipid compounds and fumonisins contamination has already been considered [36]. A positive correlation was obtained between the fatty acids, oleic, and linoleic acid when the fumonisins contamination level was higher. However, the accumulation of fatty acids in the presence of a fumonisin non-producing *F. verticillioides* strain after significant growth has never been considered. FBs were first detected by high-performance liquid chromatography-mass spectrometry (HPLC–MS/MS) after 4 dai and reached a maximum at 14 dai (Figure 3). In general, the number of lipids in infected seedlings was significantly altered compared to the control seedlings, regardless of the presence of the FBs (Appendix A). Specifically, differences between the wild-type and *fum1*∆ infected seedlings depend on the length of infection. FA C16:1 and, in general, the C18 FAs undergo a substantial change at 4 dai—they decreased in the wild-type-infected seedlings compared to the *fum1*∆-infected ones (Appendix A). At 14 dai, C16:1, C18:0, and C18:2 FA significantly increased in the wild-type-infected seedling, even in comparison with the control and, intriguingly with the *fum1*∆-infected seedlings (Figure 3). The presence of FBs is positively correlated to the increase of FAs in maize; these results are consistent with previous ones [37]. 

The influence of FB synthesis on oxylipins production was also examined (Appendix A). The oxylipin 9-HODE, an established mycotoxin-susceptibility factor [38], is produced by maize ears contaminated with fumonisins under field conditions [37]. Here we found 9-HODE concomitant with FB biosynthesis (i.e., 7–14 dai; Appendix A). We also report significative differences in seedlings infected with the naïve or the *fum1*∆ strains in relation to the synthesis of the JA precursors, 13-HOTrE, 13-HpODE, and 13-HODE. Notably, these oxylipins were enhanced after growth of the fumonisin production strain in comparison to both fumonisin non-producing strain and the uninfected control. It was recently noted that jasmonates play a leading role in controlling *F. verticillioides* infection in maize [35,37,39]. Other oxylipins, such as 10-HpOME (by C18:1), 9,10-DiHOME (by C18:2), 9-oxoODE (by C18:2), and 9-oxoOTrE (by C18:3), which seem to be linked to the precocious FBs accumulation, were detected at 7 dai, at least for the oxylipins derived from C18:2 and C18:3, as already suggested elsewhere [38,40].

### 2.3. Maize and F. verticillioides Sphingolipidome Characterization

The relationship between sphingolipids and fumonisins was tested by HPLC–MS/MS. To check and characterize the sphingolipids, a mass spectrometry method was conducted to evaluate LCBs, ceramides, phytoceramides, and dehydro-phytoceramides present in our pathosystem. The generated mass-spectra were analyzed by multiple reaction monitoring (MRM) approach. The identified compounds are listed and grouped in Appendix A according to whether they were detected in maize, in *F. verticillioides,* or in both. The analysis was performed in electrospray ionization source (ESI) positive ion mode. In general, the most abundant ion for sphingoid bases is the di-dehydrate ion [M+H^+^-2H_2_O] while the parental or molecular ion [M+H^+^] is less abundant (Figure 4). Saturated C18 LCBs, such as sphinganine and phytosphingosine, exhibited a typical 60 m/z fragment indicative of a trimethyl–ammonium moiety. Ceramides and phytoceramides exhibited a fragmentation pattern consistent with the loss of a single water moiety and include a cognate sphingoid base fragment as an amide bounded fatty acid ion. Glucosyl-ceramides harbor an additional glucose moiety that constitutes a neutral loss.

Appendix A is an overlay of total ion chromatograms from mock-infected maize and maize infected with *F. verticillioides*. Significant differences in the profiles were readily noted with ions of m/z 506, 680, 710, 736, 754, and 822 detected only in the infected maize while ions of m/z 536, 562, 564, 566, 596, 696, 698, 724, 770, 780, and 808 detected only in mock-infected maize.

### 2.4. Sphingolipidome Characterization in Maize Kernels Infected with F. verticillioides Wild-Type and Fumonisin Non-Producing (fum1∆) Mutant Strain

As noted, FBs, which *F. verticillioides* may produce while infecting maize, can inhibit the enzyme CerS and disrupt plant sphingolipid synthesis. Therefore, we reasoned that comparing lipids present in maize infected with an FB non-producing strain (*fum1*∆) and a producing strain (wild-type) would reveal the affected lipid species (Figure 3b–d). HPLC–MS/MS and statistical analysis of maize seedlings mock-infected (ctr) and infected with the wild-type or *fum1*∆ strains revealed that the presence and quantity of 13 sphingolipids were significantly different during the infection process (Appendix A). Maximal differences were observed at 14 dai (Figure 5). Sphingosine d18:1 has already demonstrated closely linked to PCD *Arabidopsis* [21], and our result in maize confirm this (Figure 5). Dihydroxy and trihydroxy LCFA containing sphingolipids (S-LCFAs) d18:0/16:0 (Figure 5), d18:2/18:1 (Figure 5), t18:0/16:0 (Figure 5), t18:1/16:0 (Figure 5) are similar in maize infected with the *fum1*∆ strain or mock-infected and significantly greater in the wild-type-infected maize at 14 dai, and the VLCFA sphingolipid (S-VLCFA) t18:0/24:0 (Figure 5). The presence of FB synthesis can positively act on the synthesis of these sphingolipids. S-VLCFAs t18:2/22:0 (Figure 5), t18:0/26:0 (Figure 5), t18:0/h23:0 (Figure 5), and t18:0/h24:0 (Figure 5) increase in the maize kernel infected with *fum1*∆ strain, suggesting that they are impacted by the presence of FBs, inhibiting the site of ceramide synthases. 

Only three sphingolipids were found to be present both in wild-type and *fum1*∆ maize infected kernel: the ceramide d18:2/22:1 (Figure 5), the phytoceramides t18:0/18:1 (Figure 5), and t18:0/h25:0 (Figure 5).

### 2.5. The Balance between the Salicylic and Jasmonic Acid

SA and JA are two crucial phytohormones involved in stress signaling and plant defense. In general, JA trigger and modulate plant resistance against biotrophic pathogens while the SA does so against the necrotrophic pathogens. Their balance and the influence of other hormones drive the defense response. In the cross-talk among SA and JA, several molecular mediators interact that drive the signal transduction. The interactions among these factors determine the plant’s defense against specific aggressors. 

SA and JA seem to have a synergistic or antagonistic interaction in plant disease and immune responses. Here we noted that SA accumulation started early after infection and continued through 14 dai (Figure 6a), while JA was present only at 7 dai and 14 dai (Figure 6b). However, the amount of JA and SA only became significant at 14 dai, when they were greater in wild-type-infected seedlings than they were in either the ctr and *fum1*∆-infected seedlings. These results suggest that the concentration of JA and SA is dependent on the increase in fumonisins and that the synergic accumulation at 14 dai may be linked to the necrotrophic lifestyle. 

### 2.6. Exposure of Maize Protoplasts to FB1

The PCD consists in the accumulation of antimicrobial compounds and in the formation of necrotic lesions to ensure pathogen containment. In addition, the SA induces apoptosis [41]. Maize protoplasts were exposed to a mixture of FB1 and FB2; after exposure, apoptotic and necrotic protoplasts have been counted. The percentage of apoptotic and necrotic cells was enhanced after the exposure (Table 1).

## 3. Discussion

*F. verticillioides* is most often a harmless endophyte of maize, but, in response to poorly understood biotic and abiotic factors, it can become a pathogen and derive nutrients as a destructive necrotroph [12]. Infected kernels, showing no or minimal disease may contain low levels of mycelia, while significantly diseased kernels are extensively colonized and are frequently contaminated with fumonisins (FBs), a family of mycotoxins that may cause significant harm to animals. What is the function of these chemicals and are they related to maize disease? Although FBs are not required for endophytic growth [42] nor for some ear-rot diseases [43], we and many others have found that they do play a role in seedling diseases of maize [11,13,20,41,42,44].

Lipid entities present during the maize kernel infection are correlated to fumonisin contamination [39]. Here we observed a drastic increase of FBs and a significant increase of C16:1, C18:0, and C18:2 fatty acids (FAs) in the wild type-infected seedlings after 14 days infection (dai) (Figure 3b–d). In addition to their role in the onset of plant defenses [45], FAs serve as precursors for oxylipins. Numerous oxylipins function as signaling molecules and are intimately involved in the maize–*F. verticillioides* communication [36]. In our system, the mycotoxin-inducing oxylipin 9-H(P)ODE is triggered upon fungal infection regardless of the ability of the fungus to produce (wild-type) or not (*fum1*∆) fumonisins. Thus, we can infer and confirm its role as a mycotoxin regulator at least in this system. Most intriguingly, the wild-type strain stimulated synthesis of the plant hormone jasmonatic acid (JA) in maize seedlings better than the *fum1*∆ strain. Based on this, it can be suggested that FBs can act as elicitors or enhancers of the plant hormones, SA, and JA (Figure 6).

Amongst lipid entities involved in the plant–fungal interaction, sphingolipids also play a crucial role. We found that FB synthesis and maize disease severity were connected by alterations in plant sphingolipids caused by the FB induced disruption of plant ceramide synthase activity. What are these changes? In *Arabidopsis*, FBs increase long-chain base (LCBs or sphingoid bases) levels, which induce PCD. FBs also disrupt sphingolipid metabolism in maize seedling, and the toxin increase causes an increase of free sphingoid bases [46]. Specific lipids, involved in triggering PCD as a consequence of FB phytotoxic effects, have not been well characterized to date. 

To help clarify the relationship in maize between sphingolipids, fumonisins, and disease, we evaluated the relative abundance of more than 50 sphingolipids (Appendix A), including different subtypes of sphingoid bases, ceramides, and phytoceramides, produced by both the host and the pathogen up to 14 dai (Appendix A and Figure 5). We found (i) an increase of sphingoid base d18:0 in wild-type-infected maize, (ii) more S-LCFA sphingolipids, such as d18:0/16:0, d18:2/18:1, t18:0/16:0, t18:2/18:1, and t18:1/16:0, and one S-VLCFA t18:0/24:0 at 14 dai in wild-type-infected maize compared to mock-infected maize; and (iii) more S-VLCFAs, such as t18:0/22:0, t18:0/26:0, t18:0/h23:0,and t18:0/h24:0 in maize infected with the non-fumonisin producing mutant *fum1*∆ compared to maize infected with the wild-type. The alterations caused by FBs on lipid metabolism generated an accumulation at 14 dai in the wild-type-infected kernel of the ceramides and phytoceramides containing LCFA. 

What are the modulators of sphingolipids in plants? In *Arabidopsis thaliana*, three isoforms (*LOH1, LOH2*, and *LOH3*) are present. The ceramide synthase 2 (*LOH2*) preferentially acylates sphinganine with C16 fatty acids, while ceramide synthase 1 and 3 (*LOH1* and *LOH3*) acylate phytosphingosine with VLCFA [47,48]. The accumulation of S-LCFA in the wild-type-infected kernel and the accumulation of S-VLCFAs in maize infected with *F. verticillioides fum1*∆ suggests that FBs produced by the fungus could target maize homologs of *LOH3*/*CerS2* over other isoforms of CerS, thereby reducing the S-VLCFA synthesis and *LOH2* to enhance the amount of S-LCFA. What is the resulting phenotype of these alterations in the sphingolipidome in our system? Probably, it is an increase of necrotic areas in the roots of the maize seedlings infected with the FB-producer, i.e., the wild-type strain. Supporting this hypothesis, FB that was provided to protoplasts obtained from our system enhanced the apoptotic/necrotrophic events, as also verified by other authors in *Arabidopsis* [22]. Thus, we can hypothesize that FB-driven massive increase of sphingoid bases and S-LCFA and a concomitant decrease of S-VLCFA represent key signals that can control PCD in maize seedlings. Sphingolipid-regulated PCD in plant cells has long been studied, but the biochemical mechanism of its regulation remains largely unclear [6].

No maize genotypes resistant or immune to *Fusarium* infection and FB contamination have been identified so far [49]. Recently, some plant gene networks were shown to reduce the negative effect of *F. verticillioides* or FBs on plant health; notwithstanding the fact that a high number of markers of resistance was identified, the small effect of each marker on disease severity is consistent with the quantitative nature of the *F. verticillioides*-maize pathosystem [50]. Here we explored the impact of FBs on the production of plant hormones SA and JA, which are two key effectors of plant defense (Figure 6a,b). In general, the SA-mediated defenses act against pathogens with a biotrophic lifestyle, whereas JA-mediated defenses act against necrotrophic pathogens. From our results, we suggest that the balance between SA and JA is fundamental during the infection progress because the increase or decrease of these hormones could influence the success of the pathogen. SA was detected through days 2–14 of the infection (Figure 6a), while JA was detected through days 7–14 of our infection assay (Figure 6b). These results show the timing of hormone accumulation correlate with the transition from a biotrophic growth to a necrotrophic lifestyle for *F. verticillioides*. SA pathway is generally known to be antagonistic to JA pathway, and they turn on/off in response to the lifestyle of the pathogen present. To contain biotrophic growth in general, SA induces an oxidative burst and activates defense-related genes PR1, PR2, and PR5, as well as genes involved in the flavonoid synthesis (e.g. PAL) [51]. Because *F. verticillioides* is considered a hemibiotrophic pathogen, we suggest that the activation of the JA signaling pathway follows the activation of SA-mediated pathways. The JA pathway promotes the expression of JAs biosynthesis-related genes such as the lipoxygenase (LOX), allene oxide synthase (AOS), allene oxide cyclase (AOC), OPDA reductase (oxo-phytodienoic acid - OPR3) [52]. Here we found that the presence of the FBs quantitatively influenced hormone production; in absence of this mycotoxin (as shown in the mutant-infection assay) the synthesis of JA and SA in seedlings was halved in the final step of infection (14 dai). 

Taken together, the data support the hypothesis that fumonisins produced by *F. verticillioides* hijack lipid (sphingolipid) metabolism of maize seedlings, which, in turn, affect host defense by impacting hormone synthesis (SA/JA), driving the cells toward the PCD. Similarly, the effector ToxA produced by *Parastagonospora nodorum*’s elicits PCD in wheat leaves, which also likely aids the switch from an epiphytic to a necrotrophic growth [53]. Thus, by comparison, fumonisins could also be effectors that likely support the necrotrophic stage of *F. verticillioides* during growth in maize.

## 4. Materials and Methods 

### 4.1. Fusarium verticillioides Strains and Culture Conditions

Fungal strains used in this study were *Fusarium verticillioides* 7600 (ATCC^®^ MYA-4922TM) supplied by American Type Culture Collection (ATCC, Manassas, VA, USA) and the *fum1* deletion mutant (*fum1*∆), which was derived from 7600. *Fum1* encodes the polyketide synthase responsible for synthesizing the fumonisin carbon backbone [54]. The mutant strain was supplied from the MycoBank of Agricultural Research Service (US Department of Agriculture, Peoria, IL, USA). For general spore production, strains were cultured on potato dextrose broth medium (PDB, Difco, Waltham, MA, USA). Liquid media was inoculated with a suspension of 1 × 10^6^ conidia per mL and incubated at 25 °C in the dark on a rotary shaker. Mycelia were harvested five days post-inoculation, filtered, lyophilized, and ground to a powder in liquid nitrogen using a mortar and pestle. 

### 4.2. Kernel Infection Assay

The kernel infection assay was performed as described in Covarelli et al. (2012) [55], with some modifications. Maize kernels (maize hybrid KXB6567 distributed by KWS Italia S.p.A.) were surface sterilized by immersion for 10 min in a 1% of sodium hypochlorite solution. Seeds were then rinsed two times for 10 min with sterile water and incubated for four hours with sterile water to cover them. The kernels were then pierced with a needle in the embryo area and immersed for 30 min in sterile water containing 10^6^ microconidia per mL and placed on filter paper saturated with sterile water in Petri dishes. Two Petri dishes containing 10 seeds each were used for each strain and mock-inoculated seeds were used as the control. The kernels (mock and infected) were incubated for 2, 4, 7, and 14 days at which point the number of necrotic roots and the length of the roots and shoots were determined. The assay was repeated in triplicate.

### 4.3. HPLC–MS/MS Analysis

All solvents used for sphingolipids extraction and HPLC/MS analysis were of HPLC/MS grade. Methanol (MeOH), isopropyl alcohol (iPrOH), ethyl acetate (EtOAc), and acetonitrile (ACN) were purchased from Merck (Darmstadt, Germany). Ammonium formate (NH_4_HCO_2_) was purchased from Sigma Aldrich.

Mass spectrometry analyses were performed by LC (HPLC 1200 series rapid resolution) coupled to a triple quadrupole MS (G6420 series triple quadrupole, QqQ; Agilent Technologies, Santa Clara, CA, USA), with an electrospray ionization source (ESI). Chromatographic column and analysis software were provided by Agilent Technologies (Santa Clara, CA, USA). The temperature was set at 350 °C, the nitrogen flow at 10 L/min, the nebulization pressure at 20 psi, and the voltage at 4000 V.

#### 4.3.1. Sphingolipids, Oxylipin, and Fatty Acid Extraction and Analysis

Lipids were extracted from 30 mg of ground, lyophilized germinated maize kernels, as reported in Ludovici et al. (2014) [44], with some modifications. The internal reference standard added for sphingolipid analysis was the C16:0-d31 ceramide N-palmitoyl-D-erythro-sphingosine (Avanti Polar Lipids, Alabaster, AL, USA), while for fatty acid and oxylipin analysis was the oxylipin 9(S)-HODE-d4 (Cayman, Ann Arbor, MI, USA) at the final concentration of 1 µM. The samples were mixed with 2 mL of extraction mixture (isopropyl alcohol: water: ethyl acetate 1:1:3 *v/v*, with 0.0025% w/v of butylated hydroxytoluene to prevent peroxidation) and then centrifuged at 12.000 rpm. The upper phase was collected and dried under a stream of nitrogen gas. The sample was then extracted with 1 mL of ethyl acetate. After centrifugation at 12.000 rpm, the upper phase was transferred to the collection tube with the first extract and dried under a stream of nitrogen gas. The combined extracts were dissolved in 100 µL of methanol and analyzed by HPLC–MS/MS.

Chromatographic separation of sphingolipids was performed with a Zorbax SB-C8 rapid resolution HT 2.1 × 50 mm 1.8 µm 600 bar column (Agilent Technologies, Santa Clara, CA, USA). The mobile phases consisted of A phase (water: B phase 40:60 *v/v*, containing ammonium formate 5 mM), B phase (methanol: acetonitrile: isopropyl alcohol 70:20:10 *v/v*). The elution program was as follows: 0–6 min 20% B, 6–14 min 98% B, 14–18 min 98% B, 18–20 min 100% B, 18–20 min 100% B, 20–22 min 100% B, 22–26 min 20% B, and 26–28 min 20% B. The flow rate was 0.6 mL/min. The column was warmed at 60 °C. The injection volume was 10 µL. HPLC–MS/MS method to identify the molecular structures of sphingolipids from biological sources was previously introduced by Sugawara et al. (2010) [56]. Sphingolipids were identified using multiple reaction monitoring (MRM) (Appendix A). 

The chromatographic separation of oxylipins and fatty acids was performed with a Zorbax ECLIPSE XDB-C18 rapid resolution HT 4.6 × 50 mm 1.8 µm column (Agilent technologies, Santa Clara, CA, USA. The mobile phase consisted of A: water/acetonitrile 97:3 *v/v* containing 0.1% formic acid and 3% acetonitrile, and B: acetonitrile/isopropyl alcohol 90:10 *v/v*. The elution program was: 0–2 min 80% A and 20% B, 2–4 min 65% A 35% B, 4–6 min 60% A and 40% B, 6–7 min 58% A and 42%, 7–9 min 52% A and 48% B, 9–15 min 35% A and 65% B, 15–17 min 25% A and 75% B, 17–18.50 min 15%A and 85% B, 18.50–19.50 min 5% A and 95% B, 19.50–24 min 5% A and 95% B, 24–26 min 1% A and 99% B, 26–30 min 1% A and 99% B, 30–34 min 80% A and 20% B. The flow rate was: 0–24 min at 0.6 mL/min, 24–30 min at 1 mL/min, 30–34 min at 0.6 mL/min. The column temperature was set at 50 °C with an injection volume of 10 µL. Oxylipins present were identified using MRM [44], while fatty acids were identified using the single ion monitoring (SIM) approach (Appendix A).

#### 4.3.2. Fumonisin Extraction and Analysis

FBs were extracted from *F. verticillioides* infected and non-infected maize kernels. A total of 500 mg of lyophilized seedling roots and shoots was ground in presence of liquid nitrogen with a mortar and pestle. The internal standard [^13^C_34_] fumonisin B_1_ (Romer Labs, Getzersrdorf, Austria) was added at the final concentration of 2 µM. FBs were extracted with a method described by Agilent Technologies (Santa Clara, CA, USA) with some modification [57]. A total of 2 mL of acetonitrile: water: acetic acid (70:29:1, *v/v/v*) was added to the ground plant material, vortexed for 1 min, and placed on a rotary shaker for 30 min. The solution was centrifuged at 7.000 rpm for 15 min. The supernatant was collected and dried under oxygen gas flux. The dried samples were resuspended in 100 µL of acetonitrile. 

Chromatographic separation of fumonisins was performed with a Zorbax ECLIPSE XDB-C18 rapid resolution HT 4.6 × 50 mm 1.8 µm column (Agilent Technologies, Santa Clara, CA, USA) and was used at 25 °C, and the injected volume was 5 µL. The mobile phases consisted of A: water: methanol: acetic acid (89:10:1, *v/v/v*) containing 5 mM of ammonium acetate, and B: methanol: water: acetic acid (2:97:1, *v/v/v*) containing 5 mM of ammonium acetate. The elution gradient was as follows: 0–2 min 1% B, 3–14 min 99% B, 15–18 min 99% B, 19–20 min 1% B. The gradient was followed by 2 min for re-equilibration. The flowrate was constant at 0,6 mL/min. The main transition and qualifier ions for FB_1_ is 722.2→352.1, for FB_2_ is 706.4→336.3, and for the internal standard [13C_34_] FB_1_ is 756.4→374.4. Fragmentor was set at 180 V and the collision energy at 45 eV. Quantitation of FB_1_ and FB_2_ in maize samples was calculated using a linear calibration curve covering a concentration range of 0.025–50 ng/mL. The equation for FB_1_ was y = 515, 27x + 47,246 (R^2^ = 0.9951) and the equation for FB_2_ was y = 5465, 9x + 72,908 (R^2^ = 0.9959). 

#### 4.3.3. SA and JA Extraction and Analysis

SA was extracted from *F. verticillioides* infected and mock-infected maize seedlings, essentially as described in Antiga et al. (2020) [58]. The quantification was carried out by the addition of internal standard 1-naphthaleneacetic acid (NAA), at a final concentration of 5 µM. A total of 30 mg of lyophilized, ground mycelium was extracted with 750 μL of methanol: water: acetic acid (90:9:1, *v/v/v*). Extraction was repeated and the dried samples were dissolved in 200 μL of a mixture of water and acetic acid (0.05%). Chromatographic separation was carried out with a Zorbax ECLIPSE XDB-C18 rapid resolution HT 4.6 × 50 mm 1.8 µm p.s. column (Agilent Technologies, Santa Clara, CA, USA), and the column was thermostated at 25°C and 10 μL were injected. The mobile phases consisted of A: water containing 0.05% acetic acid, and B: acetonitrile. The elution gradient was as follows: 0–3 min 15% B, 3–5 min 100% B, 5–6 min 100% B, 6–7 min 15% B, and 7–8 min 15% B. The gradient was followed by 5 min of the original mobile phase for re-equilibration. The flow-rate was constant at 0.6 mL/min. SA was analyzed in negative ion mode [M + H]^−^. The main transition for SA was 137.2→92.9 and the main transition for JA was 209.2→ 59.9. The internal standard NAA was analyzed in negative ion mode as the acetate adduct [M + CH_3_COOH^−^]^−^. The main transition for NAA was 245→180.8. For SA, the fragmentor was kept at 135 V and the collision energy was 20 eV, and for NAA, the fragmentor was kept at 100 V and the collision energy was 16 eV. SA and JA were identified using MRM in negative ion mode [M − H]^−^.

### 4.4. Nucleic Acid Manipulation: DNA for qPCR Analysis

For qPCR, lyophilized germinated seedlings were ground in a blender with liquid nitrogen. Total genomic DNAs were extracted from 30 mg of material employing the 3-CTAB method as previously described [59]. DNA concentration was assessed with the Nanodrop spectrophotometer (Thermo Fisher Scientific, Waltham, MA, USA) and confirmed by agarose gel electrophoresis. qPCR assays were performed using the primer pairs specific to *F. verticillioides* β-tubulin and *Z. mays* β-tubulin (Table 2). Amplifications were performed in 10 μL of SensiFAST™ SYBR^®^ No-ROX Kit (Bioline, Memphis, TN, USA) containing 200 ng of DNA. DNA quantification was determined using the 2^−ΔΔct^ method, i.e., by normalizing the *F. verticillioides* β-tubulin cycle threshold (Ct) value with the *Z. mays* β-tubulin Ct and each value of the infected samples with their relative controls [60]. 

### 4.5. Vybrant Assay for PCD Detection

Fresh maize seeds were surface sterilized by immersion in 1% of sodium hypochlorite for 10 min and rinsed two times for 10 min with sterile water. Seeds were allowed to germinate for 4 days, as described above. To release protoplasts, 1 gm of germinated maize kernels from each treatment was suspended in a 30 mL solution containing the cell wall degrading enzymes Extralyse^®^ (1.5%), Cellulase (Merck KGaA, Darmstadt, Germany) (0.5%) and Driselase (0.3%). A total of 10^6^ protoplasts were treated for 30 min with a mixture of 25 ng of FB_1_ and FB_2_ and were analyzed with the Vybrant apoptosis assay (Thermo Fisher Scientific, Waltham, MA, USA). Viable, necrotic, and apoptotic protoplasts present in the same sample were distinguished from each other by evaluating differences in cell membrane permeability and were counted using the epifluorescence motorized microscope Zeiss Imager M2 controlled by Zen pro (Carl Zeiss, Aalen, Germany) [61].

### 4.6. Statistical Analysis

Error bars represent the standard error calculated from independent replicates, technically repeated in triplicate. Rstudio was used as the statistic package. Datasets are compared using one- or two-way ANOVA followed by Tukey’s test. Other datasets were compared using Student’s t-test.

## 5. Conclusions

Our analyses indicate fumonisin as a virulence factor, demonstrating that during the infection process the accumulation of FB generates an alteration of lipid metabolism.

This study demonstrated that both FAs and S-LCFA increase in presence of FBs, causing the activation of the defense directing the seedlings toward PCD, as underlined by the presence of sphingosine d18:1 [21]. The increase of hormone synthesis (SA and JA) indicates the infection progress, suggesting the fundamental role of these molecules.

Moreover, the seedlings infected by wild-type strain are affected in the S-VLCFA accumulation [62]. Taken together, our results provided a basis for better understanding the lipid markers of PCD caused by the mycotoxigenic fungus *F. verticillioides* (Figure 7). 

## Figures and Tables

**Figure 1 ijms-22-02435-f001:**
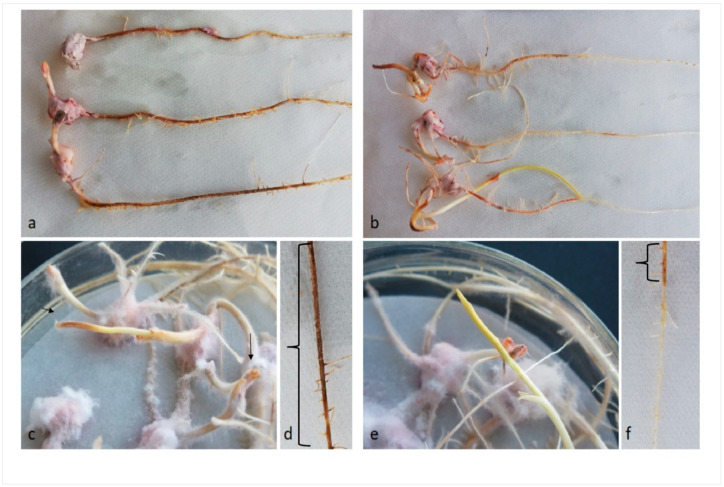
Seedling shoots and roots showing necrotic areas caused by infection with *Fusarium verticillioides* wild-type (wt) or *fum1* deletion mutant (*fum1*∆) strains. (**a**) wt and (**b**) *fum1*∆ infected seedlings at 14 days after infection (dai); close view of typical shoots and the necrotic portion of the roots in wt-(**c**,**d**) and in *fum1*∆ infected seedlings (**e**,**f**) with (the necrotic areas are indicated by the arrows—in the shoots—and between brackets—in the roots).

**Figure 2 ijms-22-02435-f002:**
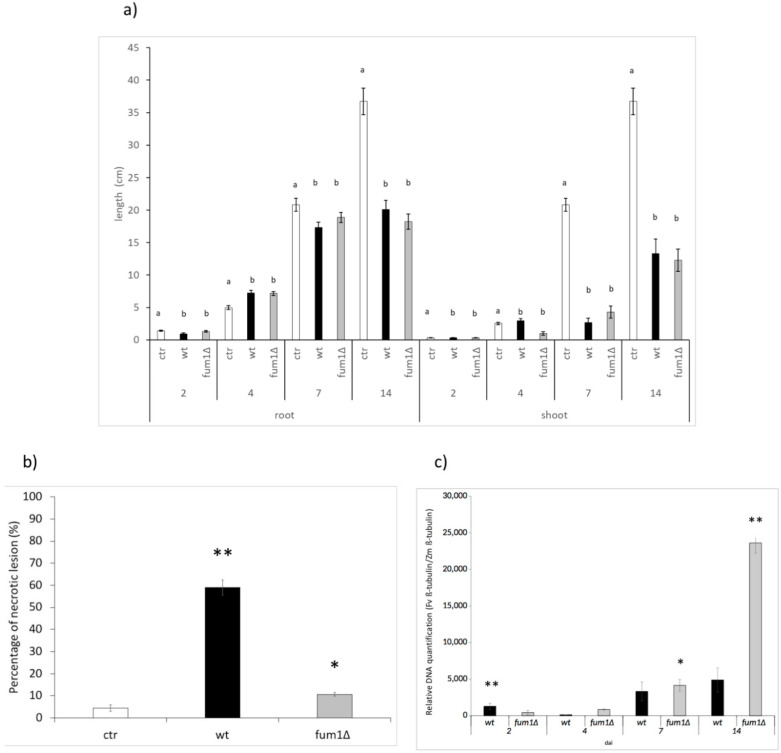
(**a**) Root and stem elongation in maize kernels mock-infected (ctr) or infected with *F. verticillioides* wild-type (wt) or mutant (*fum1*∆) after 2, 4, 7, and 14 days after infection (dai). Data are presented as the mean from two experiments, n = 20 seeds, results are the mean ± std error. Significance (*p* ≤ 0.05) was assessed by one-way ANOVA followed by Tukey’s test (groupings indicated by a,b). (**b**) number of roots showings symptoms of necrosis at 14 dai in the non-infected control (ctr), wt, and *fum1*∆ infected seedlings. (**c**) DNA extracted from infected and non-infected seedlings at 2, 4, 7, and 14 dai was examined by qPCR with primers specific to *F. verticillioides* and *Z. mays* genes coding for β-tubulin. Total height of each bar indicates the quantity of fungal DNA relative to the total DNA detected. Asterisks indicate a significant difference at same time point from ctr in (**b**) and from the wt in (**c**) (Student’s t-test: * *p* < 0.05; ** *p* < 0.01).

**Figure 3 ijms-22-02435-f003:**
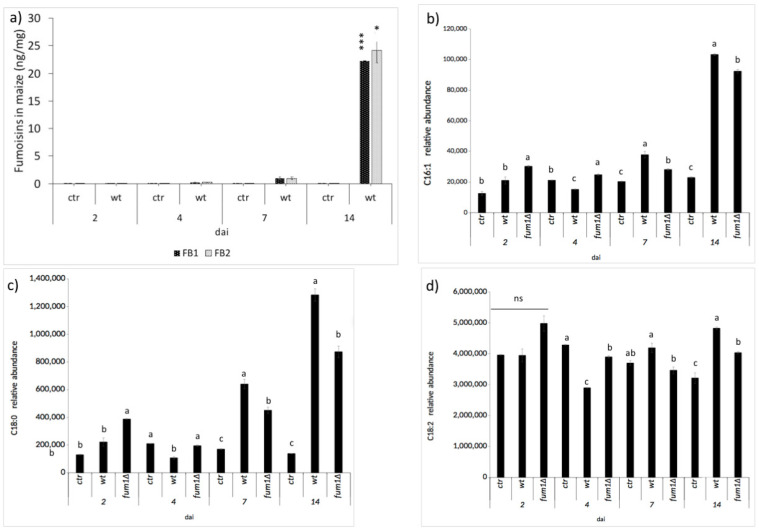
Toxin and fatty acid content were evaluated in seedlings non-infected (ctr) and infected by wild-type (wt), or by *fum1* deletion mutant (*fum1*∆). (**a**) Fumonisins (FB1 and FB2) in wt-infected and in the ctr seedlings at several days after infection (dai) analyzed by an MRM (Multiple Reaction Monitoring) approach with HPLC–MS/MS. The error bars represent standard error calculated from three independent replicates, technically repeated in triplicate. Asterisks indicate a significant difference at same time point from ctr (Student’s *t*-test: * *p* < 0.05; *** *p* < 0.001). (**b**) C16:1 (**c**) C18:0 (**d**) C18:2 fatty acids in infected (wt or *fum1*∆) and ctr seedlings at several dai analyzed by a single ion monitoring (SIM) with HPLC–MS/MS. The error bars represent standard error calculated from three independent replicates, technically repeated in triplicate. Significance (*p* ≤ 0.05) was assessed by one-way ANOVA followed by Tukey’s test.

**Figure 4 ijms-22-02435-f004:**
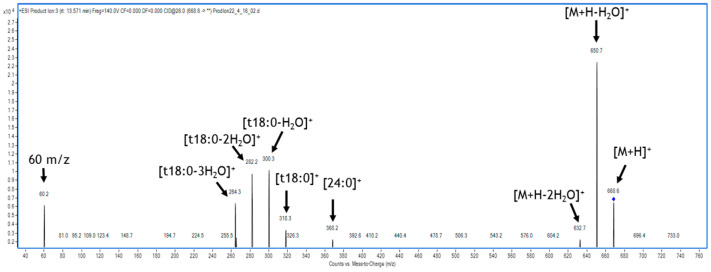
Example of fragmentation pattern for a sphingolipid. The phytoceramide with m/z 667.6 (t18:0) showed a clear fragmentation pattern with the 60 m/z fragments typical of sphingoid bases.

**Figure 5 ijms-22-02435-f005:**
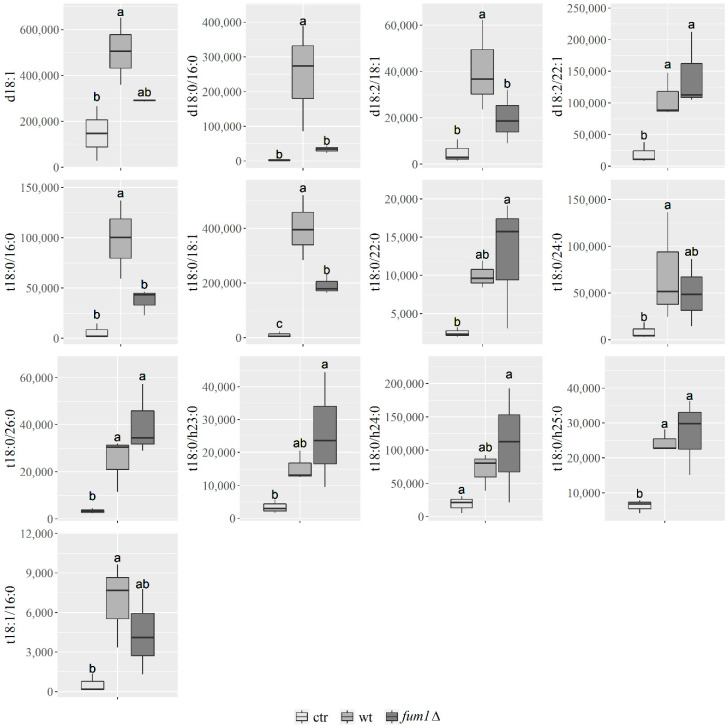
The relative abundance of different sphingolipids at 14 dai. Y-axis: relative abundance of compounds found in maize kernel mock-infected (ctr) and infected with *F. verticillioides* wild-type (wt) or fumonisin non-producing mutant *fum1*∆, measured as normalized peak intensity. Box plot represents the distribution of the relative quantity of sphingolipids deriving from three independent replicates which were technically repeated in triplicate. Asterisks indicate statistically significant differences between infected-maize (wild-type or *fum1*∆) and ctr at each time point. Significance (*p* ≤ 0.05) was assessed by one-way ANOVA followed by Tukey’s test.

**Figure 6 ijms-22-02435-f006:**
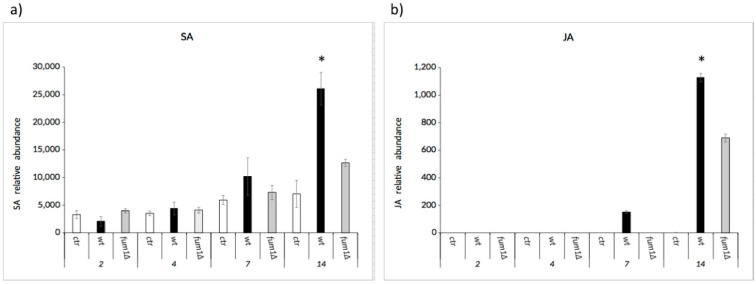
HPLC-MS/MS quantification of hormones. SA (**a**) and JA (**b**) in seedling infected by *F. verticillioides* wild-type (wt) or *fum1*∆ strains compared to seedling not infected (ctr). Data represent the means ± standard error of three technical replicates. Asterisks above the bars indicate a significant difference between the ctr and wt or *fum1*∆ infected seedlings at the same time point (Student’s *t*-test, * *p* < 0.05).

**Figure 7 ijms-22-02435-f007:**
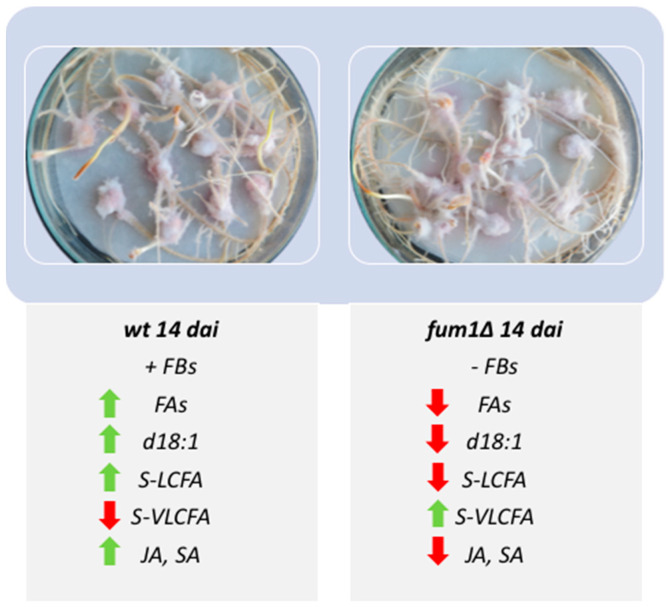
Seedlings infected at 14 dai with *F. verticillioides* wild-type (wt) or *fum1* deletion mutant (*fum1*∆) strains. Lipids, hormones, and genes are accompanied by the green or red arrow to indicate the relative increase or decrease.

**Table 1 ijms-22-02435-t001:** Cell viability of protoplasts in the absence (Ctr) or presence of 25 ng of FBs (FB_1_ + FB_2_) Values represent the mean of triplicate experiments.

	% Protoplasts					
Treatment	Normal		Apoptotic		Necrotic	
ctr	41		31		29	
Maize Protoplasts + (FB_1_ + FB_2_)	29	***	36	**	35	*

Student’s *t*-test, * *p* < 0.05, ** *p* < 0.01, *** *p* < 0.001 applied to the compare maize protoplasts + (FB_1_ + FB_2_) to non-infected seed control (ctr).

**Table 2 ijms-22-02435-t002:** Primers used in this study.

Gene	Accession No.	t Annealing (°C)	Sequence
*β-tubulin*	NP_001105457	60	Fw CTACCTCACGGCATCTGCTATGT Rev GTCACACACACTCGACTTCACG
*β-tubulin*	FVEG_04081	40	Fw CTCTGCTCATTTCCAAGATCCGCG Rev GTAGTTGAGGTCACCGTAGGAGG

## Data Availability

Data is contained within the article or Appendix A.

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
