# Peer review of "The Effect of Fusarium verticillioides Fumonisins on Fatty Acids, Sphingolipids, and Oxylipins in Maize Germlings"

_ijms, 2021, doi:10.3390/ijms22052435_

Round 1
Reviewer 1 Report
Line 44: what do you mean by FB mutant, a non-producing fumonisin strain?
Line 62, 63: references in text should be indicated by number
Figure 1: explain abbreviations (ctr, wt, etc.)
Figure 2: explain abbreviations
Explain all abbreviations in text the first time they appear, though a list in provided at the end of article
Line 364: what is PDB?
Line 425: provide the reference of the Agilent method for the analysis of fumonisins
Author Response
Thanks for your valuable comments. Here there are our answers:
Line 44: what do you mean by FB mutant, a non-producing fumonisin strain?
Correction done
Line 62, 63: references in text should be indicated by number
Correction done.
Figure 1: explain abbreviations (ctr, wt, etc.)
Correction done.
Figure 2: explain abbreviation
Correction done.
Explain all abbreviations in text the first time they appear, though a list in provided at the end of article
Checked and corrected.
Line 364: what is PDB?
Potato Dextrose Broth, added.
Line 425: provide the reference of the Agilent method for the analysis of fumonisins
Reference added.
Reviewer 2 Report
The authors Beccacciolo et al. presented a very important problem of the role of fumonisins in the pathogenicity of Fusarium strains as a key substance potentially capable of switching the endophyte into a pathogen and inducing plant resistance. Studies on lipid changes (fatty acids, oxylipins and sphingolipids) have been based on solid knowledge of the effect of fumonisin on the activity of ceramide synthase and the mechanism of action and biosynthesis pathways of these compounds. It is particularly important to link the effects of fumonisins to the process of resistance induction and the production of signaling substances created in separate but interrelated immunity pathways: jasmic acid (JA) and salicylic acid (SA). It would be interesting to determine the activity of phenylalanine lyase and pathogen-releted proteins or at least refer to this problem in the introduction and discussion.
Well-chosen research tools were used and maize seeding trials were properly prepared for testing.
However, the presentation of the results leaves much to be desired. It is difficult to compare the changes in the composition of sphingolipids presented on Fig. 3. The graphs should be presented in a much more compact form for easy comparison, namely Fig. 3 should be reduced and occupy a maximum of 3/4 pages. Subtitles on this Figure are almost impossible to see. The quality of all Figures should be improved. Add in Figs and Tables uniform letter markings of significance determined by ANOVA. A significant part of the results should be presented in the main work and not in the supplement. It is necessary to move Fig.S1 as well as Fig. S4 and Table S7 to the main part of paper.
It is a pity that the authors did not estimate the abundance of F. verticilloides or biomass of this fungus in the inoculated tissues of Wt and fum1? Or maybe only these results have not been presented? Macroscopic photos of infection do not allow you to compare the degree of paralysis.
Authors should unify the mock/control record.
The literature of the last 3 years should also be supplemented.
Author Response
Thank you for your suggestions, the work has been revised in all them parts.
Below the answers to your comments.
- The authors Beccaccioli et al. presented a very important problem of the role of fumonisins in the pathogenicity of Fusarium strains as a key substance potentially capable of switching the endophyte into a pathogen and inducing plant resistance. Studies on lipid changes (fatty acids, oxylipins and sphingolipids) have been based on solid knowledge of the effect of fumonisin on the activity of ceramide synthase and the mechanism of action and biosynthesis pathways of these compounds. It is particularly important to link the effects of fumonisins to the process of resistance induction and the production of signaling substances created in separate but interrelated immunity pathways: jasmic acid (JA) and salicylic acid (SA). It would be interesting to determine the activity of phenylalanine lyase and pathogen-releted proteins or at least refer to this problem in the introduction and discussion.
the “theme” suggested by the reviewer has been now included in introduction (67-75) and discussion (548-555).
- However, the presentation of the results leaves much to be desired. It is difficult to compare the changes in the composition of sphingolipids presented on Fig. 3. The graphs should be presented in a much more compact form for easy comparison, namely Fig. 3 should be reduced and occupy a maximum of 3/4 pages. Subtitles on this Figure are almost impossible to see. The quality of all Figures should be improved. Add in Figs and Tables uniform letter markings of significance determined by ANOVA. A significant part of the results should be presented in the main work and not in the supplement. It is necessary to move Fig.S1 as well as Fig. S4 and Table S7 to the main part of paper.
The presentation of the results was improved, in particular for the figure 3 (now Fig. 5) included the subtitle.
The quality of all figures has been implemented.
All the figures and table present the letters to assess the significance determined by ANOVA.
Fig.S1 (now Fig.2a) , Fig. S4 (now Fig.4) and Table S7(now table 2) are now part of the main text.
- It is a pity that the authors did not estimate the abundance of verticilloides or biomass of this fungus in the inoculated tissues of Wt and fum1? Or maybe only these results have not been presented? Macroscopic photos of infection do not allow you to compare the degree of paralysis.
The abundance of F. verticillioides was estimated by a quantitative molecular approach, as reported in (now) Fig. 2c we estimated the quantity of DNA.
- Authors should unify the mock/control record.
Control record has been uniformed in the text and in the figs.
- The literature of the last 3 years should also be supplemented.
Literature has been implemented in particular REF 21, 60, 69, 73.
Reviewer 3 Report
In this article, the authors suggest that during the interaction between the host and F. verticillioides, the modification of sphingolipid and plant defence responses are associated with the fumonisins production and modulate the endophytic growth of F. verticillioides into a necrotrophic growth typical of rotting diseases.
It is an interesting study, which is well done and well written.
In my opinion the paper can be accepted after minor modifications:
References should be checked:
- References should include DOI numbers.
- The name of the organisms should be written in italics (ref. 3, 6, 10, 11, 12, 13, 14, 20, 22, 25, 28, 34, 37, 41, 42, 43, 45, 46, 48, 49, 53, 56, 57, 62, 63, 64, 65, 66, 67, 70, 71), and some references need to be standardized in the title, for example, ref. 7, 14, 23 and 56 (initials in lowercase letters).
- Some references are uncompleted: ref. 2, 3, 6, 7, 8, 9, 15, 16, 17, 22, 23, 24, 42 and 54.
In summary, this paper includes so useful information but requires minor modifications to make it suitable for publication.
Author Response
Thank you for your kind suggestions. Please find our answers below:
In this article, the authors suggest that during the interaction between the host and F. verticillioides, the modification of sphingolipid and plant defence responses are associated with the fumonisins production and modulate the endophytic growth of F. verticillioides into a necrotrophic growth typical of rotting diseases.
It is an interesting study, which is well done and well written.
In my opinion the paper can be accepted after minor modifications
References should be checked:
- References should include DOI numbers.
References are formatted by the style required by the journal (International Journal of Molecular Science)
- The name of the organisms should be written in italics (ref. 3, 6, 10, 11, 12, 13, 14, 20, 22, 25, 28, 34, 37, 41, 42, 43, 45, 46, 48, 49, 53, 56, 57, 62, 63, 64, 65, 66, 67, 70, 71), and some references need to be standardized in the title, for example, ref. 7, 14, 23 and 56 (initials in lowercase letters).
Checked and corrected.
- Some references are uncompleted: ref. 2, 3, 6, 7, 8, 9, 15, 16, 17, 22, 23, 24, 42 and 54.
Checked and corrected.
In summary, this paper includes so useful information but requires minor modifications to make it suitable for publication.
Reviewer 4 Report
The paper provides a valuable, detailed account of the effect of fumonisins on metabolic responses of maize seedlings that are important in understanding mechanism of phytotoxicity of fumonisins. I have no criticisms of the way in which the work was done, and I do not question the results. As far as interpretation, I would like to point out that some of the sphingolipids that accumulate due to inhibition of ceramide synthase are phytotoxic, causing similar symptoms to FBs (see Tanaka, T. et al. 1993. Structure-dependent phytotoxicity of fumonisins and related compounds in a duckweed bioassay. Phytochemistry 33: 779-785). The concentrations of these compounds can be much higher than the concentration of FB that causes their accumulation. The last figure of this paper shows that FB1 and phytosphingosine as synergistic in causing cellular leakage (plasma membrane dysfunction).
Some minor suggestions:
A better title might be:
The effects of Fusarium verticilliodes fumonisins on fatty acids, sphingolipids, and oxylipins in maize seedlings
Line #
23 – fumonisin-producing
24 – fumonisin-deficient
47 – synthesize
48 - …13], and ….
54 – forms
62, 63 - …PCD (20, 21).
93 – FB-producing
97 - ..can explain…’
101- ..with the FB production…
Author Response
we modify the text according to your kind suggestions